# The Influence of Harvest Moment and Cultivar on Variability of Some Chemical Constituents and Antiradical Activity of Dehydrated Chokeberry Pomace

Ivona Enescu Mazilu [1,2] , Loredana Elena Vîjan [3,*] and Sina Cosmulescu [4,*]

1 Doctoral School of Plant and Animal Resources Engineering, Faculty of Horticulture, University of Craiova, 13 A.I. Cuza Street, 200585 Craiova, Romania; icmazilu@yahoo.com
2 Research Institute for Fruit Growing Pitesti, 402 Mărului Street, 117450 Mărăcineni, Romania
3 Faculty of Sciences, Physical Education and Computer Science, University of Pitești, 1 Târgu din Vale Street, 110040 Pitesti, Romania
4 Department of Horticulture and Food Science, Faculty of Horticulture, University of Craiova, 13 A.I. Cuza Street, 200585 Craiova, Romania
* Correspondence: loredana.vijan@upit.ro (L.E.V.); sinacosmulescu@hotmail.com (S.C.)

**Abstract:** This paper's aim was to study how the antioxidant activity and the level of certain phenolic complexes and carotenoids vary in the pomace obtained from the fruits of two cultivars of chokeberry at different times of harvest after reaching the stage of maturity. The influence of the cultivar, harvest moment, and the combined effect of these two factors on the antioxidant activity and the dehydrated pomace content in components with antioxidant potentials, such as total phenolics, total tannins, total flavonoids, lycopene, and β-carotene was analyzed. The methanolic extract from the pomace obtained from the 'Melrom' cultivar had the highest efficiency ($92.14 \pm 5.02\%$). The antiradical activity of the pomace was maximal ($93.27 \pm 4.32\%$) after the middle of the harvest season (3 September). The pomace obtained from the 'Nero' cultivar displayed superior levels of phenolic content ($13,030.16 \pm 1414.46$ mg/100 g), flavonoids ($4627.83 \pm 509.63$ mg CE/100 g), tannins ($7458.56 \pm 529.43$ mg/100 g), and lycopene ($1.171 \pm 0.388$ mg/100 g). The 'Melrom' cultivar presented superior content of β-carotene ($0.313 \pm 0.07$ mg/100 g). On average, a positive significant correlation between radical scavenging activity with total phenolic content and β-carotene was observed. The combined cultivar × harvest moment effect was reflected in the variations in the total tannins content and the total flavonoid content, but also in the antiradical activity of the methanolic extracts. Dehydrated pomace from chokeberry fruit can be an important source of antioxidant biological compounds and can be used to make innovative foods.

**Keywords:** antioxidant activity; β-carotene; flavonoids; lycopene; phenolics; tannins

## 1. Introduction

Adopting a healthy lifestyle is based, among other factors, on the concept of quality diet, which does not only satisfy the basic need for food but is also considered to contribute to the maintenance of a healthy lifestyle or to even enhance it. Hence, the concept of healthy food has gained popularity among consumers and producers and has brought new challenges to the scientific community. A series of defense systems against free radicals is part of the human organism, which contains antioxidant enzymes, such as catalase, peroxidase, and superoxide dismutase, as well as antioxidants such as ascorbic acid and tocopherol [1]. The activity of antioxidants consists of inhibiting or delaying the oxidation process of susceptible molecules even at significantly reduced concentrations compared to the substrate [2]. Under certain improper conditions (intense stress), the organism's defense systems may be overpowered, which drives the need to supplement it with dietary antioxidants, of which phenolic compounds represent one of the main

classes of antioxidants contained in plants. Nevertheless, in the vegetal kingdom, there are also other types of compounds that possess antioxidant properties, such as vitamin C, carotenoids, and tocopherols, that can act per se or can be associated, forming complexes with higher efficiency [3]. Berries and more specifically those from the *Rosaceae* and *Ericaceae* families represent sources of biologically active compounds such as fibers (dietetical) and antioxidant complexes [4–7]. One member of the *Rosaceae* family, *Aronia melanocarpa* (Michx) Ell., has been included in the group of intensely studied species, due to its small, dark purple colored fruits' remarkably high content of antioxidants, vitamins, and mineral elements [8–13]. In addition, *A. melanocarpa* has shown the characteristics of a well-adapted and producing crop in one study area [14] and presented a wide harvest season [13,15]. The astringent and sometimes bitter taste of chokeberry justifies the interest in processing it, even at minimal levels, to obtain juice (combined with other fruits or not), jams, liquors, etc. [8,16]. Another way to exploit chokeberry benefits is the introduction of mashes or powders obtained from dehydrated fruits into pastry or dairy products [17–22]. A series of studies [2,23] has shown that the epicarp of fruits represents an antioxidant source. Moreover, according to Ahmed et al. [24], the epicarp of fruits represents a richer antioxidant source compared to their mesocarp and, through the pressing process of chokeberry fruits, the pomace, rich in active principles, is obtained [25]. Additionally, it is appreciated that the management of by-products in the fruit processing industry is a major issue in this field, both from an economic and environmental perspective [26]. Hence, it is necessary to process the residual and exploit it. A quick solution that also extends the storage period is dehydrating pomace and using it as an ingredient in the food industry. At the same time, pomace can also represent a valuable source of recovering the complexes with biological activity. The biological value of the vegetal processed materials depends on the parameters of thermic treatments applied and the storage conditions [27,28]. Furthermore, processing technologies that ensure the retention of larger quantities of desired complexes have been developed, such as non-thermic processing (PEF, HP, ultrasonication, and irradiation) [29]. A simple approach for the quick and inexpensive processing of chokeberry fruits, or other fruit species, is low-temperature dehydration. Considering that the chemical composition of fruits and therefore of post-processing residuals changes depending on the orchard technology or environmental factors, this paper proposes an analysis of how the antioxidant activity and the levels of certain phenolic complexes and carotenoids do vary in the pomace obtained from the fruits of two varieties of chokeberry at different times of harvest after reaching the stage of maturity. In addition, considering the advantage of a crop with a wide harvest season, the proper choice of harvest moment could be a useful tool to achieve chokeberry by-products with the highest biologically active content.

## 2. Materials and Methods

### 2.1. Chemicals and Reagents

Methanol, 2,2-diphenyl-1-picrylhydrazyl (DPPH), sodium hydroxide, sodium carbonate, sodium nitrite, aluminum chloride, acetone, n-hexane, ethanol, gallic acid, catechin, and Folin–Ciocalteu (Merck, Darmstadt, Germany) were used in this experiment.

### 2.2. Material

The material was represented by the fruits of two cultivars of *Aronia melanocarpa* (Michx.) Elliot, 'Melrom' and 'Nero', grown in the experimental field of the Research Institute for Fruit Growing, Pitesti-Maracineni (44°54′11″ N 24°52′29″ E, 287 m elevation a.s.l.). An automated weather station, WatchDog900ET (Spectrum Technologies Inc., Aurora, IL, USA), positioned near the experimental plots, was used for weather data monitoring. The climate of the area is humid temperate-continental, with a multiannual (1969–2019) average temperature of 10.0 °C, 679.1 mm rainfall, and 75% humidity. Over the period of October 2020–September 2021, the mean temperature was 11.1 °C and the rainfall was 665.8 mm, while there was a total of 2196.2 sunshine hours. The winter temperature ranged between 0.5 and 6.2 °C in December, between −3.3 °C and 5.3 °C in January, and between −2.0 °C

and 9.1 °C in February. The spring temperature varied between −1.3 °C and 10.5 °C in March, between 2.6 °C and 15.0 °C in April, and between 9.0 °C and 22.3 °C in May. During the summer months, the temperature rose to 26.6 °C in June, 31.1 °C in July, and 31.1 °C in August. The extreme values of the temperature dropped to −7.9 °C in December, −14.1 °C in January, −10.3 °C in February, and −6.2 °C in March, and rose to 22.3 °C in February, 18.8 °C in March, 25.3 °C in April, 28.4 °C in May, 34.0 °C in June, 36.8 °C in July, and 36.4 °C in August. The lowest air humidity was 47.7% (April), and air humidity under 70% was registered in January, February, March, May, June, and July. In addition, total sunshine hours reached a maximum of 306.7 h in July and ranged between 243.7 and 296.6 h in April, May, June, and August. The plantations were established in 2011 ('Nero') and 2017 ('Melrom'). In 2021, a completely randomized design experiment with two factors was set up, in which the first factor was the harvest moment and the second factor was the cultivar. The experimental units were arranged randomly in three replications (100 plants per replication) for each cultivar. Because the fruits reached their maturity on 13 August, the first harvest moment was not considered in the experiment. Therefore, the fruits were harvested after reaching their full maturity (BBCH 87), in 7 stages, every 5 days in 2021, from 13 August to 13 September. The freshly harvested fruits (20 kg samples) were washed with potable water, dried with a paper towel, and pressed with an industrial Voran Hydraulic press 100 P2 series, with a pressing force of 24 t for obtaining the juice. The pressed residual (pulp, skin, seeds) resulting from the juice production was convectively dehydrated at 45 °C for 24 h with an industrial Hestya DRY 15 electrical dehydrator until the water content of 10% was obtained. Next, samples of 200 g each of dehydrated residual were finely grained with a coffee grinder and stored in glass containers hermetically closed at 4 degrees Celsius, in a dark place. The Vortex Mixer VX-200 Corning-Labnet system, ultrasonic bath with heater, cleaner ULTR-2L0-001 and the centrifuge Labnet spectralfuge 6c were used.

### 2.3. Obtaining the Methanolic Extract

A total of 0.5 g of vegetal dehydrated and ground material was treated with 10 mL methanol-water (8:2, *v/v*) and vortexed for 2 min at 3000 rpm. Next, the mix was introduced for 20 min into an ultrasonic bath (40 kHz). After the 15 min centrifuge at 3000 rpm, the supernatant was separated through filtering using a Whatman No. 1 filter and used to determine the total phenolic content and total flavonoid content. The same procedure was followed for obtaining the methanolic extract (using methanol 99.8%) needed for determining the antiradical activity.

### 2.4. Obtaining the Aqueous Extract

The extract needed for the TTC dosing was obtained by treating 0.5 g dehydrated and ground chokeberry pomace with 100 mL ultrapure water followed by the samples' vortex treatment for 2 min at 3000 rpm and by the samples' incubation in the ultrasonic bath at 80 °C (40 kHz, 100 W) for 30 min. The aqueous extract was represented by the supernatant extracted after the centrifuge of the samples for 15 min at 3000 rpm.

### 2.5. Determining the Components with Antiradical Potential

For all the colorimetrical measurements, a spectrophotometer UV-Vis Perkin Elmer Lambda 25 was used.

### 2.6. Determining the Total Phenolic Content

Quantitative determination of polyphenols was performed according to the methodology proposed by Matić et al. [30] and Cosmulescu et al. [31]. The principle of the method is based on forming a blue-colored compound between phosphotungstic acid and polyphenols, in an alkaline medium. For analysis, 0.5 mL methanolic extract was added to 5 mL ultrapure water and 0.5 mL Folin Ciocalteu reagent mixture. After 10 min, 2 mL 10% sodium carbonate was added, and the mixture was completed to a final volume with 10 mL

ultrapure water. The absorbance readings were taken at 760 nm after 2 h of incubation at room temperature, in dark conditions. Gallic acid was used as a reference standard, and the results were expressed as milligrams of gallic acid equivalent (mg GAE)/100 g dry weight of vegetal material.

### 2.7. Determining the Total Tannins Content

The methodology followed by Giura et al. [32], with some modifications was used. For analysis, 1 mL aqueous extract was added to a 10 mL flask containing 2 mL of distilled water and 2 mL of Folin-Ciocalteu reagent. After 5 min of rest, a 5 mL solution of sodium carbonate 10% was added. After 60 min of rest, the absorbance of the samples was measured at 760 nm and the concentration of tannins was expressed as mg GAE/100 g dry weight of vegetal material.

### 2.8. Determining the Total Flavonoids Content

For the quantification of flavonoids, the methodology proposed by Matić and Jakobek et al. [33] and Cosmulescu et al. [34] was used. The principle of the method is based on the formation of a yellow-orange-colored compound by the reaction of flavonoids and aluminum chloride. For analysis, 1 mL of methanolic extract was added to a 10 mL volumetric flask containing 4 mL of distilled water and 0.3 mL of sodium nitrite 5%. After 5 min of rest, 0.3 mL of aluminum chloride 10% was added to the volumetric flask. After 5 min, a 2 mL solution of sodium hydroxide 1 M was added and was diluted with distilled water up to the final volume of 10 mL. The solution absorbance was measured at 510 nm. The total flavonoid content was expressed as mg catechin equivalent (CE)/100 g dry weight of vegetal material.

### 2.9. Determining the Lycopene and β-Carotene Levels

Quantitative determination of carotenoids (lycopene and β-carotene) was performed based on the methodology proposed by Tudor-Radu et al. [35]. For extraction of the two compounds, 1 g of dehydrated chokeberry pomace was used, which was added to a 25 mL mixture of solvents (hexane: ethanol: acetone in a 2:1:1 volume ratio). The mixture was stirred for 30 min at 1500 rpm and then 10 mL of distilled water was added. Then, stirring was continued for another 10 min. After 15 min of rest, the phases were separated. The concentration of carotenoids was expressed as mg/100 g vegetal material, and calculated using molar extinction coefficients of 184,900/M cm at 470 nm and 172,000/M cm at 503 nm for lycopene [36], while 108,427/M cm at 470 nm and 24,686/M cm at 503 nm was used for β-carotene in hexane [35].

### 2.10. Determining the Radical Scavenging Activity

The DPPH test was used to evaluate the radical scavenging activity (RSA) of the studied methanolic extracts. The method was based on the observation that, when the DPPH radical was scavenged by an antioxidant through the donation of hydrogen to form reduced DPPH-H, the color of the DPPH solution turned from purple to yellow and the molar absorptivity at 517 nm decreased. A solution of DPPH in methanol ($1.16 \times 10^{-4}$ mol/L) was prepared, and 2.965 mL of this solution was mixed with 0.035 mL methanolic extract (containing $1.75 \times 10^{-3}$ g powder of dehydrated pomace). The reaction mixtures were vortexed thoroughly and left to stand in the dark at room temperature for up to 20 min. The absorbance of the mixture was spectrophotometrically measured at $\lambda_{max}$ = 517 nm. The percentage of DPPH radical scavenging activity (RSA%) of extracts was calculated based on the formula: y = [1 − (sample absorbance/blank absorbance)] × 100 [34,37], where blank absorbance is the molar absorptivity at 517 nm of a mixture containing 2.965 mL DPPH methanolic solution ($1.16 \times 10^{-4}$ mol/L) and 0.035 mL methanol 99.8%, prepared as above. Methanol was used for the baseline correction.

### 2.11. Statistical Analysis

All analyses were performed in triplicate and data were reported as mean ± standard deviation (SD), except for RSA, where data were presented as mean ± SD of four determinations. Results were processed by Excel (Microsoft Office 2010) and SPSS Trial Version 28.0 (SPSS Inc., Chicago, IL, USA). Data were subjected to analysis of variance (two-way ANOVA; $p \leq 0.05$), and Duncan's Multiple Range Test (DMRT) post hoc tests were used to measure specific differences between sample means.

### 3. Results and Discussion

The results for certain chemical constituents and the antiradical activity of Aronia dehydrated pomace, depending on the variety and the time of harvest, are presented in Tables 1–5. As presented in Table 1, the total phenolic content (TPC) of the dehydrated pomace recorded an average value of 12,747.9 mg GAE/100 g DW, oscillating between 10,112.1 for the 'Melrom' cultivar at the beginning of the harvesting period (13 August), and 15,288.38 mg GAE/100 g DW for 'Nero', at the end of this period (13 September). The best-represented class of phenolic compounds from the dehydrated chokeberry fruit pomace was the tannins class (TTC), representing approximately 57.43% of the total phenolic content, while flavonoids (TFC) accounted for 31.49%. Similar to the total phenolic content, tannins oscillated depending on the harvest date, with a medium level of 7243.4 mg GAE/100 g DW obtained. Therefore, tannins reached a maximum value for the 'Melrom' cultivar in the middle of the harvesting season (9092.5 mg GAE/100 g DW), and the smallest tannins content (5219.1 mg GAE/100 g DW) was obtained for 'Melrom', at the beginning of the harvest. Flavonoids accumulated in the chokeberry fruit as the harvest was delayed, and their highest level was recorded in the pomace obtained from the fruits harvested during the second half of the season, on 3 September, for the 'Nero' cultivar (5244.3 mg CE/100 g DW).

**Table 1.** Mean values, standard deviation, variation coefficients, and limit values of total phenolic content (TPC), total tannins content (TTC), total flavonoids content (TFC), lycopene, β-carotene, and radical scavenging activity (RSA) for the dehydrated chokeberry pomace.

| Statistics | TPC (mg GAE/100 g) | TTC (mg GAE/100 g) | TFC (mg CE/100 g) | Lycopene (mg/100 g) | β-Carotene (mg/100 g) | RSA % |
|---|---|---|---|---|---|---|
| Mean | 12747.93 | 7243.42 | 3971.86 | 1.04 | 0.30 | 88.84 |
| Std. dev. | 1348.12 | 850.74 | 777.07 | 0.35 | 0.08 | 8.14 |
| CV% | 10.5 | 11.7 | 19.5 | 33.90 | 26.50 | 9.17 |
| Minimum | 10,112.17 | 5219.1 | 2660.8 | 0.58 | 0.18 | 59.62 |
| Maximum | 15,288.3 | 9092.5 | 5244.4 | 2.04 | 0.45 | 98.68 |
| p cultivar | <0.001 | <0.001 | <0.001 | 0.007 | n.s.* | <0.001 |
| p harvest period | <0.001 | <0.001 | <0.001 | 0.004 | n.s. | 0.017 |
| p cultivar × harvest period | n.s. | <0.001 | <0.001 | n.s. | n.s. | 0.021 |

* n.s.= non-significant ($p > 0.05$).

**Table 2.** Variations in total phenolic content (TPC) and total flavonoid content (TFC) in dehydrated chokeberry pomace during the harvest season *.

| Harvest Period | TPC (mg GAE/100 g) | | | TFC (mg CE/100 g) | | |
|---|---|---|---|---|---|---|
| | 'Melrom' | 'Nero' | Average | 'Melrom' | 'Nero' | Average |
| 13th Aug. | 10,240.8 ± 128.4 [d] | 10,467.0 ± 272.9 [e] | 10,353.9 ± 227.5 [d] | 2927.3 ± 336.7 [c] | 4004.8 ± 113.1 [c] | 3466.0 ± 631.4 [bc] |
| 18th Aug. | 11,232.9 ± 398.0 [c] | 12,224.6 ± 601.4 [d] | 11,728.8 ± 709.3 [c] | 3236.02 ± 142.4 [bc] | 3963.8 ± 150.1 [c] | 3599.9 ± 419.6 [bc] |
| 23th Aug. | 12,587.7 ± 522.6 [b] | 12,818.1 ± 493.7 [cd] | 12,702.9 ± 471.9 [b] | 3267.1 ± 139.7 [bc] | 4295.9 ± 182.6 [b] | 3781.5 ± 581.9 [b] |
| 28th Aug. | 13,206.4 ± 337.2 [ab] | 13,190.1 ± 611.0 [c] | 13,198.3 ± 441.4 [b] | 3348.9 ± 85.5 [b] | 4932.7 ± 211.0 [a] | 4140.8 ± 879.3 [a] |
| 3rd Sep. | 13,632.7 ± 205.1 [a] | 14,126.0 ± 270.3 [b] | 13,879.4 ± 345.1 [a] | 3274.8 ± 119.8 [bc] | 5131.4 ± 124.7 [a] | 4203.1 ± 1022.7 [a] |
| 8th Sep. | 12,816.0 ± 407.2 [b] | 13,425.7 ± 470.4 [bc] | 13,120.8 ± 516.1 [b] | 3708.0 ± 170.9 [a] | 4992.0 ± 184.1 [a] | 4350.0 ± 720.9 [a] |
| 13th Sep. | 13,543.2 ± 537.6 [a] | 14,959.2 ± 327.4 [a] | 14,251.0 ± 1350.5 [a] | 3448.8 ± 200.4 [ab] | 5073.8 ± 91.8 [a] | 4261.3 ± 900.9 [a] |
| Average | 12,465.7 ± 1248.1 | 13,030.1 ± 1414.4 | | 3315.9 ± 272.9 | 4627.8 ± 509.6 | |

* Means with the same letter are not significantly different at 5% level.

**Table 3.** Variations in total tannins content (TTC) and the lycopene content in dehydrated chokeberry pomace during the harvest season *.

| Harvest Period | TTC (mg GAE/100 g) | | | Lycopene (mg/100 g) | | |
|---|---|---|---|---|---|---|
| | 'Melrom' | 'Nero' | Average | 'Melrom' | 'Nero' | Average |
| 13th Aug. | 5475.4 ± 242.4 [e] | 6970.9 ± 184.9 [c] | 6223.1 ± 841.5 [d] | 0.603 ± 0.02 [b] | 0.880 ± 0.11 [b] | 0.741 ± 0.18 [c] |
| 18th Aug. | 6298.8 ± 145.3 [d] | 8206.1 ± 365.3 [a] | 7252.4 ± 1073.8 [b] | 0.938 ± 0.07 [b] | 1.081 ± 0.19 [ab] | 1.009 ± 0.15 [bc] |
| 23th Aug. | 7035.2 ± 248.7 [c] | 7771.9 ± 270.9 [ab] | 7403.6 ± 465.7 [b] | 0.822 ± 0.56 [b] | 0.965 ± 0.18 [ab] | 0.893 ± 0.14 [bc] |
| 28th Aug. | 8896.7 ± 227.1 [a] | 7807.4 ± 295.9 [ab] | 8352.1 ± 641.6 [a] | 0.867 ± 0.14 [b] | 0.908 ± 0.18 [ab] | 0.887 ± 0.15 [bc] |
| 3rd Sep. | 6850.8 ± 250.6 [c] | 7550.0 ± 146.5 [b] | 7200.4 ± 424.6 [b] | 0.967 ± 0.10 [ab] | 1.419 ± 0.46 [ab] | 1.193 ± 0.39 [ab] |
| 8th Sep. | 6850.5 ± 259.1 [c] | 6884.7 ± 317.2 [c] | 6867.6 ± 259.7 [c] | 0.895 ± 0.16 [b] | 1.384 ± 0.57 [ab] | 1.140 ± 0.46 [ab] |
| 13th Sep. | 7790.2 ± 348.0 [b] | 7018.6 ± 174.4 [c] | 7404.4 ± 489.1 [b] | 1.329 ± 0.48 [a] | 1.558 ± 0.36 [a] | 1.444 ± 0.40 [a] |
| Average | 7028.2 ± 1051.7 | 7458.5 ± 569.4 | | 0.917 ± 0.27 | 1.171 ± 0.38 | |

* Means with the same letter are not significantly different at 5% level.

**Table 4.** Variations in β-carotene content and the radical scavenging activity (anti-DPPH) of dehydrated chokeberry pomace during the harvest season *.

| Harvest Period | β- Carotene (mg/100 g) | | | RSA (%) | | |
|---|---|---|---|---|---|---|
| | 'Melrom' | 'Nero' | Average | 'Melrom' | 'Nero' | Average |
| 13th Aug. | 0.230 ± 0.02 [a] | 0.237 ± 0.05 [a] | 0.233 ± 0.04 [a] | 94.62 ± 1.79 [a] | 74.39 ± 12.41 [c] | 84.51 ± 13.5 [bc] |
| 18th Aug. | 0.360 ± 0.08 [a] | 0.292 ± 0.06 [a] | 0.326 ± 0.07 [a] | 90.40 ± 3.68 [a] | 77.05 ± 10.56 [bc] | 83.71 ± 10.23 [c] |
| 23th Aug. | 0.338 ± 0.08 [a] | 0.292 ± 0.08 [a] | 0.315 ± 0.07 [a] | 89.95 ± 2.21 [a] | 87.67 ± 6.62 [ab] | 88.81 ± 4.73 [abc] |
| 28th Aug. | 0.314 ± 0.06 [a] | 0.291 ± 0.07 [a] | 0.302 ± 0.06 [a] | 95.14 ± 4.12 [a] | 91.12 ± 5.35 [a] | 93.13 ± 4.91 [a] |
| 3rd Sep. | 0.335 ± 0.06 [a] | 0.329 ± 0.10 [a] | 0.332 ± 0.07 [a] | 95.89 ± 0.74 [a] | 90.66 ± 4.97 [a] | 93.27 ± 4.32 [a] |
| 8th Sep. | 0.274 ± 0.06 [a] | 0.324 ± 0.11 [a] | 0.299 ± 0.08 [a] | 90.62 ± 5.29 [a] | 90.98 ± 4.82 [a] | 90.80 ± 4.69 [ab] |
| 13th Sep. | 0.340 ± 0.12 [a] | 0.297 ± 0.12 [a] | 0.319 ± 0.11 [a] | 88.37 ± 9.40 [a] | 87.05 ± 3.65 [ab] | 87.71 ± 6.64 [abc] |
| Average | 0.313 ± 0.07 | 0.294 ± 0.08 | | 92.14 ± 5.02 | 85.56 ± 9.35 | |

* Means with the same letter are not significantly different at 5% level.

**Table 5.** The intensity of the correlations between the phenolic content (TPC, TTC, TFC), carotenoids (lycopene and β-carotene), and antiradical activity of dehydrated chokeberry powder.

| Pearson Correlation Coefficient (r) | | TPC | TFC | TTC | Lycopene | β- Carotene |
|---|---|---|---|---|---|---|
| RSA % | Pearson Correlation | 0.342 * | −0.102 | −0.054 | 0.179 | 0.526 ** |
| | Sig. (2-tailed) | 0.027 | 0.519 | 0.732 | 0.257 | 0.000 |
| TPC | Pearson Correlation | 1 | 0.568 ** | 0.487 ** | 0.567 ** | 0.356 * |
| | Sig. (2-tailed) | | 0.000 | 0.001 | 0.000 | 0.021 |
| TFC | Pearson Correlation | | 1 | 0.273 | 0.543 ** | 0.084 |
| | Sig. (2-tailed) | | | 0.080 | 0.000 | 0.598 |
| TTC | Pearson Correlation | | | 1 | 0.221 | 0.265 |
| | Sig. (2-tailed) | | | | 0.159 | 0.090 |
| Lycopene | Pearson Correlation | | | | 1 | 0.588 ** |
| | Sig. (2-tailed) | | | | | 0.000 |

\* Correlation is significant at the 0.05 level (2-tailed), \*\* Correlation is significant at the 0.01 level (2-tailed).

TFC reached its lowest level (2660.8 mg CE/100 g) at the first harvest, for the 'Melrom' cultivar. Among the carotenoids, lycopene registered an average of 1.04 mg/100 g and oscillated between the maximal level of 2.04 mg/100 g, for the 'Nero' cv. on 8 September, and its lowest limit, 0.58 mg/100 g, obtained for 'Melrom' on 13 August. The medium level of the β-carotene was 0.3 mg/100 g, while the maximal one was registered for 'Melrom' at the last harvest (0.45 mg/100 g). Compared to phenolic compounds, both the carotenoids presented higher variation coefficients (33.9% for lycopene and 26.5% for β- carotene). The medium value for radical scavenging activity was 88.84%, and oscillated from 59.62% ('Nero', 13 August) to 98.68% ('Melrom', 28 August). As Table 1 presents, significant differences between the two chokeberry cultivars were observed regarding the TPC, TTC, TFC, and lycopene levels, as well as regarding their capacity to reduce the DPPH radical. The influence of the harvest moment on TPC, TTC, TFC, and RSA was also significant. The combined effect of both the cultivar and the harvest moment was significant for TTC, TFC, and RSA.

The results regarding the influence of the harvest moment on the TPC and TFC determined in the methanolic extract of dehydrated chokeberry pomace are presented in Table 2. A tendency to accumulate mostly phenolics was observed for 'Nero', especially flavonoids, as the harvest was delayed. The accumulation of a higher level of TPC was observed earlier for 'Melrom' (from 28 August), while the postponed harvest favored the TFC and lycopene levels, but also reduced TTC (Tables 2 and 3), which could have been reflected by less bitter fruits. Tannins (Table 3) showed initially higher levels in the 28 August samples, although these levels decreased until the last harvest. β-carotene (Table 4) reached higher levels in the 'Nero' cv. in the last half of the picking interval and followed an oscillated tendency for the 'Melrom' cv. On average, 'Nero' cv. pomace presented a TPC level significantly superior to 'Melrom' cv., 13,030.1 mg/100 g (Table 2), and recorded its maximum TPC level on 13 September (14,959.2 mg/100 g), later when compared to 'Melrom' cv. (13,632.7 mg/100 g, on 3 September). Our results were in agreement with the value of 110 ± 5.6 mg GAE/100 g DW reported for TPC by Hwang et al. [38]. Lower levels of 31–63 g TPC/kg fresh pomace were determined by Mayer-Miebach et al. [39], and other lower levels were also reported in other researchers' papers: Sidor et al. [11] (63.1 g GAE/kg pomace) and Samoticha et al. [40] (4954–7265 mg GAE/100 g DM dry fruits). However, similar values were reported in studies regarding TPC analysis performed through chromatographic methods: 8044–15,058 mg/100 dry pomace [41], 15,607.48–24,447.77 g/100 g DM pomace powder (Oszmiański and Lachowicz, [42]), and 10,583.27 mg/100 g DM pomace (Oszmiański and Wojdylo [43]).

As Table 2 presents, total flavonoid content predominated in the 'Nero' cv., with an average value of 4627.8 mg CE/100 g recorded. Their maximum level, 5131.4 mg CE/100 g, was determined for the 'Nero' cv. on 3 September, and exceeded the 3708.0 mg CE/100 g level found for the 'Melrom' cv. on 8 September. The TFC results obtained in our study fall between previously reported data by Oszmiański and Wojdyło [43] and Hwang et al. [38]. Tolic et al. [9] reported for chokeberry juice a flavonoid content between 6994 and 9710 mg/L. The same authors reported an average flavonoid content of 2327 mg/100 g [9] in the dehydrated chokeberry pomace powder, while Milutinović et al. [44] reported 49.36 mg catechin/g in the ethanolic chokeberry extract, and Thi and Whang [45] reported between 52.0 and 66.1 mg CE/g flavonoids for chokeberry dried through different methods.

The 'Nero' cultivar was notable for its superior total tannins content, with an average of 7458.5 mg GAE/100 g DW obtained (Table 3). However, the maximum level of TTC found for 'Melrom', 8896.7 mg GAE/100 g DW on 28 August, exceeded the maximum TTC determined for the 'Nero' cultivar 10 days earlier (8206.1 mg GAE/100 g DW). Our results showed approximatively double TTC levels compared to Georgiev and Ludneva's [46] data reported for chokeberry purees (1.04 % immediately after processing and 1.32% after 6 months of storage). In addition, Calalb and Onica [47] found higher tannin levels, of 2–4.157%, in chokeberry fruits. In other studies, Mladin et al. [48] reported for the 'Nero' cultivar 0.733–1.174% tannins in dehydrated fruits.

The differences between lycopene levels (Table 3) were highlighted in the 'Nero' cultivar, with an average lycopene content of 1.171 mg/100 g obtained. Starting from 0.603 mg/100 g at the start of the harvest season, the lycopene accumulated until it reached 1.329 mg/100 g on 13 September for the 'Melrom' cultivar, and increased from 0.880 to 1.558 mg/100 g for 'Nero' over the same time interval. An average of 0.313 mg/100 g β-carotene (Table 4) was determined for 'Melrom' (Table 4), whereas a smaller content (0.294 mg/100 g) was found for 'Nero'. Both cultivars had the lowest β-carotene content at the beginning of the harvest season and reached their maximums on 18 August (0.360 mg/100 g, Melrom) and 3 September (0.330 mg/100 g, Nero), respectively. Literature references report smaller lycopene content values (0.6 mg/kg), although they report higher β-carotene (1.16 g/kg) values in chokeberry dehydrated fruits [49]. Nevertheless, a higher β-carotene content of 770.6 μg/100 g was reported by Tanaka and Tanaka [50] in chokeberry fruits.

Antiradical activity of methanolic dry pomace extracts varied non-significantly and in a narrower range for the 'Melrom' cultivar (88.4–95.9%, $p = 0.194$) compared to 'Nero' (74.39–91.12%, $p = 0.017$) during the harvest period (Table 4). On average, the 'Melrom' cultivar presented 7.14% higher RSA than 'Nero'. The highest differences between cultivars (Table 4) were recorded for the 13 and 18 August samples, when RSA presented 21.38 and 14.79% higher values for 'Melrom' compared with 'Nero'. Antiradical activity recorded a decreasing trend initially for 'Melrom' at the beginning of the harvest season, but reached its maximum on 3 September (95.98%), while for 'Nero', RSA increased as harvest was delayed, and its maximum was recorded five days earlier than for 'Melrom', on 28 August (91.12%). Lower values for the antioxidant potential, $83.2 \pm 3.6\%$ and $76.2 \pm 9.9\%$, respectively, were determined by Andrzejewska et al. [51] for methanolic extracts from chokeberry pulp. The comparative analysis of chokeberry antiradical activity performed by Jakobek et al. [52] positioned the 'Nero' cultivar in second place, following the wild chokeberries. Other studies [8,53] listed chokeberries as having the most antioxidant activity power, before elderberry, blueberry, and black and red currant. In addition, Milutinović et al. [44] found that the antioxidant activity of ethanolic chokeberry extract was superior when tested in vitro compared to chokeberry fruits or juice, but they were inferior when tested in vivo (BKL test). In our study, the antiradical activity of chokeberry pomace was positively correlated with TPC ($r = 0.342$) and β-carotene ($r = 0.526$), whereas TPC presented a positive correlation with TFC ($r = 0.568$), TTC ($r = 0.487$) and both lycopene ($r = 0.567$) and β-carotene ($r = 0.356$) (Table 5). Lycopene presented a positive correlation with TFC ($r = 543$) and β-carotene (0.588).

According to Gülçin et al. [54], the antioxidant activity of tannins is related to their molecular weight, i.e., the number and degree of polymerization of hydroxyl groups [55]. Carotenoids belong to the group of fat-soluble pigments, capture peroxyl radicals and act predominantly as antioxidants [56]. Similar to carotenoids, phenolics can prevent oxidative damage [57].

## 4. Conclusions

Our study, which involved harvesting mature berries at different moments, pressing them for juice extraction, and analyzing the resulted chokeberry pomace, showed that the harvest moment, similar to the cultivar, influenced significantly the composition of dehydrated pomace extracts regarding the total phenolic content, in general, and tannins and flavonoids, in particular. The study, in addition, recorded a significant fluctuation in lycopene content. The antiradical activity registered for the methanolic extract of chokeberry dry pomace was influenced particularly by the cultivar, while harvest moment had only a non-significant effect. The combined cultivar x harvest moment effect was reflected in the variations in total tannins content and total flavonoid content, but also in the antiradical activity of methanolic extracts. Presented data could be used as a tool for selecting the most suitable chokeberry cultivar and harvest moment to obtain the specific by-products for applications in the food industry. Dehydrated pomace from chokeberry fruit can be an important source of antioxidant biological compounds and can be used to make innovative foods. Further research is needed to determine the other antioxidant content in chokeberry pomace powder and the use of pomace as an additive in various food recipes.

**Author Contributions:** Conceptualization, I.E.M. and S.C.; methodology, I.E.M. and L.E.V.; validation, L.E.V.; investigation, I.E.M. and S.C.; writing—original draft preparation, I.E.M. and S.C; writing—review and editing, I.E.M. and S.C.; supervision, S.C. All authors have read and agreed to the published version of the manuscript.

**Funding:** This research received no external funding.

**Institutional Review Board Statement:** Not applicable.

**Informed Consent Statement:** Not applicable.

**Data Availability Statement:** Not applicable.

**Conflicts of Interest:** The authors declare no conflict of interest.

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
