# Peer review of "The Influence of Harvest Moment and Cultivar on Variability of Some Chemical Constituents and Antiradical Activity of Dehydrated Chokeberry Pomace"

_horticulturae, doi:10.3390/horticulturae8060544_

Round 1

Reviewer 1 Report

1) The work has many references with more than 10 years, and some with more than 5 decades. 2) The methods used in the analyzes have already been updated, so it is necessary to review mainly the references used in the methodology. 3) The work is interesting, but it did not use modern techniques of quantification and identification of compounds with antioxidant potential.

Reviewer 2 Report

The article "The influence of harvest moment and cultivar on variability of some chemical constituents and antiradical activity of dehydrated chokeberry pomace" presents an interesting subject. Has scientific relevance

It is necessary to correct some details to improve:

Instead of beta carotene put β-carotene (lines 18-19, 24, 25...)

Put in vitro in italics.

Fix word separations like posi-tive (line 24), ther-mic (line 69), irradi-ation (line 72), vari-eties (line 78).

Materials and methods - Put the other reagents because there are many missing.

Correct ml to ml (line 125, 134...)

Adjust the units by placing superscripts (example: lines 187 and 189).

In Table 1, put the meaning of all acronyms.

As a suggestion, I could do a multivariate analysis of data to observe the correlations in a more comprehensive way and suggest future solutions.

Update references as most are older than the last 5 years.

In Table 5 correct the values that are with a comma.

Round 2

Reviewer 1 Report

Os autores responderam minhas perguntas.